# A Combined AHP-PROMETHEE Approach for Portfolio Performance Comparison

Mirza Sikalo *, Almira Arnaut-Berilo and Adela Delalic

School of Economics and Business, University of Sarajevo, Trg oslobodjenja—Alija Izetbegovic 1, 71000 Sarajevo, Bosnia and Herzegovina
* Correspondence: mirza.sikalo@efsa.unsa.ba

**Abstract:** Comparing portfolio performance is complex due to the fact that each model is dominant in its own risk space. Since there is no single dominant performance measure, the research problem is how to incorporate several different measures into a performance evaluation model that allows portfolios to be ranked. In this regard, the objective of this study was to develop a new comprehensive method for comparing portfolio performance based on multiple-criteria decision-making (MCDM). This paper proposes an integrated approach for stock market decision making that combines the Analytic Hierarchy Process (AHP) and the Preference Ranking Organization Method for Enrichment Evaluations (PROMETHEE), which allow hierarchical evaluation of a finite number of alternatives according to different criteria. This hybrid approach is especially advantageous, utilizing the strengths of both individual methods. AHP enables the decomposition of a complex problem into its constituent parts and the determination of weights for criteria, while the PROMETHEE method allows the investor to determine the preference function, complete ranking, and analysis of the robustness of the results. For the MCDM model in this study, different dimensions of performance measures are considered criteria: return measures, risk measures, stability measures, and predictability measures. The methodology has been applied in comparing real portfolios selected on the basis of different risk measures. For this purpose, weekly return data were used for a sample of stocks that are components of the STOXX Europe 600 Index for the period 2000–2020. In addition, a sensitivity analysis is performed to investigate the strength of the results of this method. It suggests that the simultaneous consideration of different performance measures and the investor's attitude towards the importance of these measures are notably important in the portfolio efficiency estimation process.

**Keywords:** multiple-criteria decision-making; AHP; PROMETHEE; portfolio performance; optimization; portfolio evaluation

## 1. Introduction

Although many portfolio selection decisions are still made on a qualitative basis, in recent decades the focus has shifted to a quantitative approach to solving the portfolio selection problem (Milhomem and Pereira Dantas 2022). The first model that quantifies risk and return, as well as the relationship between them, by focusing on the correlation between returns and placing portfolio selection decision making in a formal framework based on objective parameters, was developed by Markowitz (1952). This model is considered the birth of modern portfolio theory (MPT). Since then, a number of different models have been presented to enable efficient optimization, using different risk measures for this purpose. All of these measures can be divided into two major families: deviation measures and downside risk measures (Ghahtarani et al. 2022). The question arises, however, of how to measure the success of portfolios formed on the basis of these measures. The development of models for solving optimal portfolio selection problems has been accompanied by the development of measures for evaluating portfolio performance. The relationship between the models and the techniques for measuring performance stems from the fact that the

measurement of portfolio performance is a kind of control mechanism for the portfolio selection decision and is based on a whole set of techniques developed on the basis of MPT and post-MPT. The purpose of performance measurement is to determine whether the portfolio's performance exceeds expectations and whether the performance achieved is the result of luck, knowledge, or risk-taking. In choosing a performance measurement method, a balance must be struck between the simplicity of implementation and the accuracy and understandability of the resulting information (Le Sourd 2007).

It is not easy to compare models based on different risk measures (Hunjra et al. 2020) because each model dominates its own risk space (Byrne and Lee 2004). Accordingly, comparing the performance of different models requires the use of several different indicators (Kalayci et al. 2019). The motivation for this study is the development of a comprehensive evaluation approach that reflects the multidimensionality of performance. Although the integration of AHP and PROMETHEE has been applied to other financial decisions (Ishak et al. 2019), the contribution of this study is the introduction of a new MCDM approach based on AHP and PROMETHEE for portfolio performance comparison problems.

The rest of this paper is organized as follows: Section 2 presents previous research findings related to portfolio ranking and the use of various MCDM methods in portfolio management. Section 3 describes the more theoretical framework of the AHP and PROMETHEE methods, as well as the procedure and importance of their integration in a combined approach. Section 4 presents the empirical results of the study, complemented by corresponding sensitivity analyses. Finally, Section 5 provides recommendations and concludes the paper.

## 2. An Overview of Literature

Portfolio management, i.e., stock selection and evaluation, is one of the main areas of MCDM interest (Halimi et al. 2021). Zopounidis et al. (2014) point out that MCDM methods have attractive and distinctive characteristics suitable for making financial decisions, as they allow the management of risk and uncertainty and the improvement of the traditional two-criteria trade-off between risk and return. Ponsich et al. (2013) provide a comprehensive review of the literature on multi-criteria programming for portfolio and other optimization problems. They also conclude that the use of MCDM algorithms in portfolio management is still relatively rare. Recently, several research works related to MCDM have been proposed for stock portfolio selection using different methods, such as median-scaling (MS), Technique for Order Preference by Similarity to an Ideal Solution (TOPSIS), Analytic Hierarchy Process (AHP) and additive Data Envelopment Analysis (DEA) (Pätäri et al. 2018), TODIM (por. Tomada de Decisao Interativa Multicriterio) (Alali and Tolga 2019), AHP integrated with grey relational analysis (GRA) (Nguyen et al. 2020), combined AHP-TOPSIS (Vásquez et al. 2022), Elimination and Choice Translating Reality (ELECTRE) (Palma et al. 2022), PROMETHEE (Basilio et al. 2018; Tepeli and Özkoç 2020) etc. However, Vuković et al. (2020) have shown that most methods provide similar results when using several methods for ranking selected stocks.

Most of the aforementioned papers explore the possibility of using different MCDMs to solve the problem of portfolio selection based on financial and non-financial indicators. However, MCDM is not widely used to compare and evaluate the performance of different portfolios. Some authors, such as Schaarschmidt and Schanbacher (2012) and Righi and Borenstein (2018), compare the performance of portfolios using a simple method based on average ranks, which entails ranking the obtained portfolios from best to worst for each scenario and for each metric. Martel et al. (1988) were the first to point out the possibility of comparing different risk measures due to the multidimensional nature of risk, using the ELECTRE method for a multi-criteria ranking of portfolio performance. The ELECTRE method was chosen due to its simplicity, but the authors emphasized that it is possible to use any other method, which is a matter of each investor's preference and inclination. They create fifty portfolios based on data on historical returns and compare them using four parameters: the average monthly return, variance of returns, price-to-earnings (P/E)

ratio, and net capitalization value. The authors state that other quantitative or qualitative criteria can be added to reflect the investor's personality. Bouri et al. (2002) assure that rating scale methods are appropriate for the portfolio selection problem because the ratings according to each of the criteria are on a cardinal scale and the number of relevant criteria is limited. This also shows that rating scale methods are useful not only to avoid redundancy, explained by the important interaction between different criteria, but also to satisfy the condition of readability while being aware of the exhaustion. They recommend using the PROMETHEE II method before the ELECTRE III method because it is easier to understand and perform.

Pendaraki and Zopounidis (2003) use the PROMETHE II method to solve the mutual fund performance ranking problem. They use 50 random weighting combinations for each of the seven performance measures: annual percentage change of net asset value of the mutual fund, beta coefficient, value-at-risk, annual return, Treynor index, Sharpe ratio, and Jensen's alpha coefficient. These random weighting combinations were also used, considering the absence of an expert stock market analyst who could determine the weights of the performance measures used in this study. Likewise, Sielska (2010) engages three multi-criteria outranking methods (PROMETHEE, WSA, and TOPSIS) to create rankings of mutual funds and evaluate their performance. Uygurtürk (2013) evaluates the performance of 10 Turkish pension funds using the ELECTRE method, using: the Sharpe ratio, Treynor index, Information ratio, Fama measure, and Jensen's alpha coefficient as criteria. They do not attach importance to the determination of weights and use a weight of 1/n for each criterion.

However, none of these papers compare the performance of portfolios based on different optimization models or use the AHP-PROMETHEE approach to compare portfolio performance.

## 3. Theoretical Framework

This section provides a brief description of the multi-criteria optimization methods AHP and PROMETHE II, which are analyzed in this paper. An approach combining AHP and PROMETHEE methods for ranking alternatives has already been used in various fields (Turcksin et al. 2011; Komchornrit 2021), but this is the first time that AHP and PROMETHEE have been used as a hybrid model comparing different portfolio performances.

The procedure consists of several steps. By using the AHP method, the problem, alternatives, and criteria are defined, and a hierarchical structure is formed. Since PROMETHEE does not provide any formal guidelines on how weights can be determined, the AHP addresses how to determine the weight of each criterion ($w_j$), as the most commonly used method for this purpose in hybrid MCDM models (Basilio et al. 2022). This is followed by the determination of the calculation table and preference function according to the PROMETHEE method, which uses the PROMETHEE II method for the complete ranking of alternatives. Turcksin et al. (2011) pointed out that the combination of both methods enables a careful evaluation of the identified alternatives, revealing their strengths and weaknesses, and provides a ranking that facilitates the final decision-maker's choice.

### 3.1. AHP

The Analytic Hierarchy Process (AHP) is a widely applied method used for multi-criteria decision-making that was initially developed by Saaty (1977). It is a useful approach for solving complex problems that involve subjective estimation and could be used as an assessment tool that determines the degree of importance of alternatives through pairwise comparisons. The AHP method is based on three principles: (1) the definition of hierarchy, (2) evaluation by pairwise comparison, and (3) calculation of priorities (weight scores) (Saaty 1994). In the first step, AHP enables the interactive design of the problem hierarchy as the preparation of a decision scenario and then the evaluation of pairs of elements of the hierarchy, i.e., objectives, criteria, and alternatives. After the hierarchical structure is

established, the criteria are pairwise compared with determine the weighting factors of all elements of the hierarchy.

The fundamental comparison scale for AHP is proposed by Saaty (1977). Saaty's scale is used to assess the values of the criteria weight ratio and the importance of the alternatives. That is a measuring scale that has five degrees of intensity and four intermediate degrees, where each of them corresponds to a value of judgment about how many times one criterion is more important than another (Table 1). The weights of each criterion are derived by means of pairwise comparisons in the AHP method.

**Table 1.** The Fundamental Scale for Pairwise Comparison.

| Intensity of Importance | Definition | Explanation |
|---|---|---|
| 1 | Equal importance | Two elements contribute equally to the objective |
| 3 | Moderate importance | Experience and judgment moderately favor one activity over another |
| 5 | Strong importance | Experience and judgment strongly favor one activity over another |
| 7 | Very strong importance | An activity is strongly favored and its dominance demonstrated in practice |
| 9 | Complete dominance | The evidence favoring one activity over another is of the highest possible order of affirmation |
| 2, 4, 6, 8 | Intermediate values | When compromise is needed |
| Reciprocals | If activity i has one of the above numbers assigned when compared with activity j, then j has the reciprocal value when compared with i | |

Source: Saaty (1990).

If n is the number of criteria and $a_{ij}$ is the relative importance of the i-th criterion with respect to the j-th criterion, then the pairwise comparison matrix is expressed in the form of a square matrix nxn:

$$A = \begin{bmatrix} 1 & a_{12} & \cdots & a_{1n} \\ \frac{1}{a_{12}} & 1 & \cdots & a_{2n} \\ \cdots & \cdots & \cdots & \cdots \\ \frac{1}{a_{1n}} & \frac{1}{a_{2n}} & \cdots & 1 \end{bmatrix} \tag{1}$$

In the matrix A, the vector of weights of the compared criteria (priority vector) $w_i$, $i = \overline{1, n}$ can be determined by the prioritization procedure. The additive normalization method was used to determine the criteria's priority vector. The following relations describe the given procedure:

$$S_i = \sum_{j=1}^{n} a_{ij} \tag{1}$$

$$w_i = S_i \Big/ \sum_{i=1}^{n} S_i \tag{2}$$

In practice, it happens that the matrix A contains inconsistent estimates. If it is assumed that A is a positive reciprocal matrix of order n, then $\lambda_{max}$ is the maximum eigenvalue of a matrix, for which $\lambda_{max} \geq n$ holds. When $\lambda_{max} = n$, the matrix A satisfies the full consistency property (Pant et al. 2022). Any deviation from consistency affects the change in eigenvalues, so the consistency index is defined by the AHP method as follows:

$$CI = \frac{\lambda_{max} - n}{n - 1} \tag{3}$$

Using the consistency index as a measure of the consistency of the deviation n from $\lambda_{max}$, the consistency ratio can be calculated:

$$CR = \frac{CI}{RI} \tag{4}$$

where RI is a random consistency index that depends on the value of n. To validate the judgment matrix, the critical value of CR is set at 0.10 (Saaty 1994). If CR exceeds this value, the evaluation process should be repeated to improve consistency.

Since the local weights of the alternatives are determined by prioritization, the final step of the AHP involves synthesis, which consists of adding the product of the local weights of alternatives to the weight coefficients of the associated criteria. As a final result, composite vectors of the weight coefficients of the alternatives are obtained. In this study, the alternatives are not prioritized. The application of the AHP was completed at the criteria level, and the resulting weights were used in the PROMETHEE method.

### 3.2. PROMETHEE II

PROMETHEE II belongs to the family of PROMETHEE methods developed by Brans (1982), which are methods of partial aggregation, also called outranking methods. While the PROMETHEE I method provides a partial ranking of alternatives, the basic principle of the PROMETHEE II method is that it provides the possibility of a complete ranking of alternatives. This method is well suited for problems with a finite number of alternatives that need to be ranked considering multiple, sometimes conflicting criteria (Albadvi et al. 2007). Three crucial issues in the application of this method are the selection of the generalized criteria, the determination of the criteria weights, and finally the evaluation of the parameters for each generalized criterion (Pendaraki and Zopounidis 2003). The implementation of this method requires two types of information: information about the weights of the criteria and information about the decision maker's preference function used in comparing the contributions of alternatives (Altınırmak et al. 2016).

In this study, the AHP method is used to determine the criterion weights that satisfy the condition:

$$\sum_{j=1}^{k} w_j = 1, \ j = 1, 2, \ldots, k \tag{5}$$

Then, the PROMETHEE is carried out through several steps. The first step is preference modeling, where the preference function $P_i(a, b)$ is determined for each criterion based on one of the available forms of the preference function proposed by Brans et al. (1986) (Usual, Linear, U-shape, V-shape, Level, Gaussian). The aim of this step is to judge how much alternative a is preferred over alternative b for each criterion. Let $P_i(a, \ b)$ be the preference function associated with the criterion $C_i$:

$$P_i(a, b) = F_i[C_i(a) - C_i(b)] \tag{6}$$

where $0 \leq P_i(a, b) \leq 1$, and $F_i$ is a non-decreasing function of the observed deviation between two alternatives a and b over the criterion $C_i$. With this information, an overall preference index $\pi(a, b)$ can be computed, taking all the criteria into account, which is called aggregation:

$$\pi(a, b) = \sum_{i=1}^{n} P_i(a, b) \cdot w_i \tag{7}$$

This outranking preference index determines the intensity of preference a in relation to b, taking all criteria into account. When $\pi(a, b)$ is closer to 0, the global preference for alternative a versus b is weaker. When $\pi(a, b)$ is closer to 1, the global preference for alternative a versus b is stronger. Following the same principle, $\pi(b, a)$ shows the preference for alternative b in relation to a.

The third step is the calculation of outranking flows, which is performed based on the calculation of the leaving flow ($\phi^+$) and the entering flow ($\phi^-$) for each alternative according to the following formulas:

$$\phi^+(a) = \frac{1}{n-1} \sum \pi(a, b) \tag{8}$$

$$\phi^-(a) = \frac{1}{n-1} \sum \pi(b, a) \tag{9}$$

The leaving flow (positive flow) expresses how much an alternative dominates the other alternatives, while the entering flow (negative flow) shows how much an alternative is dominated by the other alternatives. The final ranking of the alternative depends on the value of the net outranking flow:

$$\phi(a) = \phi^+ - \phi^- \tag{10}$$

The higher the value of $\phi(a)$, the better the alternative is.

## 4. Empirical Findings

The proposed methodology is applied in order to select the best performance portfolio. Decision process is structured in three main stages: Data Collection, AHP and PROMETHEE.

### 4.1. Data Collection

This study uses weekly trading data for 57 randomly selected stocks from the STOXX Europe 600 listed continuously from January 2000 to December 2020. The number of stocks in the sample follows the practice of Lee and Gankhuyag (2020) and maintains the fact that less than 60 stocks are needed to realize the full benefits of diversification, and consequently, risk is reduced to the level of market risk (Raju and Agarwalla 2021).

According to Lee and Gankhuyag (2020), special importance should be given to analysis during and after crises. Therefore, the observer period was chosen due to the fact that the European stock market experienced several periods of high volatility and instability in the financial markets caused by events such as the early 2000s recession, the global financial crisis of 2007–2009, the pandemic COVID-19, etc. This implies an additional challenge for portfolio optimization strategies, as all the models used are limited to assigning positive weights, i.e., the impossibility of short selling.

#### 4.1.1. Submitting the Alternatives

The weights of the selected stocks were assigned using models based on different risk measures that are most frequently used (Birungi and Muthoni 2021). Based on the mathematical framework of mean-variance (Markowitz 1952), mean-MAD (Konno and Yamazaki 1991), mean-CVaR (Rockafellar and Uryasev 2000), and minimax (Young 1998) models, the portfolios with the lowest risk level were selected, whereby the risk in each model was measured differently. A confidence interval of 95% was used to calculate the CVaR value. In addition, we included and tested the naïve diversification portfolio and the portfolio with the highest Sharpe ratio value. The portfolios obtained based on the optimization of these models represent alternatives in the decision-making process.

#### 4.1.2. Key Objectives Identifying and Translating into Criteria

Out-of-sample performance was evaluated by applying a "buy and hold" strategy with a portfolio holding period of one year. Since the beginning of modern portfolio theory, and especially with the development of the Capital Asset Pricing Model (CAPM) and models based on loss as a measure of risk, many measures have been proposed in the literature for the theoretical and practical evaluation of portfolio performance (Cogneau and Hübner 2009). Recently, this number has grown so large that there is no single list of all of them. Also, there is no consensus on the most appropriate measure of portfolio performance (Adcock et al. 2020). Because of the diversity of the results, the problem of selecting an attractive portfolio should include additional variables to create a more stable portfolio (Bouri et al. 2002; Pätäri et al. 2018). Then, it is a multi-criteria issue that should be tackled by using the appropriate techniques, contrary to the theoretical expectations of conventional theory, which takes into account only return and risk. Therefore, the criteria were chosen based on the following indicators: risk measures, return measures, stability measures, and predictability measures.

Average weekly return and cumulative return, which are the most frequently used measures, were used as return measures (Lee and Gankhuyag 2020). Among the risk measures, measures from the models (variance, MAD, CVaR, and maximum loss) were used so that each portfolio was in an equal position, which is in accordance with Birungi and Muthoni (2021). It is important to point out that the first two measures refer to deviation risk, and the second two measures refer to downside risk. The third group of measures, called stability measures, expresses the relationship between return and risk. Birungi and Muthoni (2021) used different return-risk ratios according to the same principle. We included the most commonly used ratios: the Sharpe ratio (Sharpe 1994), the CVaR ratio (Cogneau and Hübner 2009), the Calmar ratio (Young 1991), and the Omega ratio (Keating and Shadwick 2002). Finally, the predictability measures evaluate the deviation of the achieved returns from the expected value of the model. The smaller this difference, the more successful the model, as explained in Sikalo et al. (2022).

*4.2. AHP*

Based on the information about the objectives, the criteria, and the alternatives, a hierarchical decision tree is constructed (Figure 1).

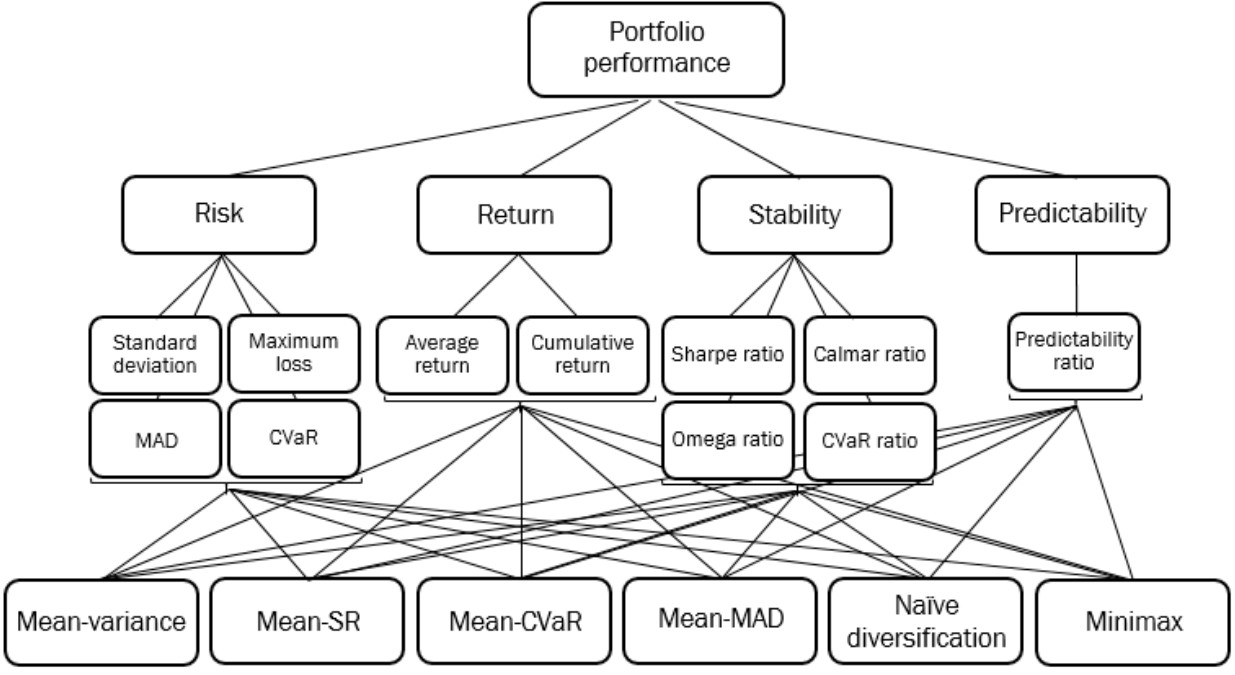

**Figure 1.** Hierarchical decision tree.

In the next step, weights were assigned to the criteria using the AHP procedure. For this purpose, specialized decision-support software Expert Choice 11 was used. In this paper, the weights are based on the judgment of a consulted group of experts with significant experience in portfolio management, following the recommendations of Mitkova and Mlynarovič (2019), Vuković et al. (2020), Vásquez et al. (2022), etc. Figure 2 shows the weighting values obtained for all criteria. Overall, return measures receive the highest preference (39.7%), followed by risk measures (25.7%), stability measures (23.4%), and prediction measures (11.2%). These weights reflect the investor's attitude towards the importance of different criteria. Sub-criteria, i.e., criteria at the next level of the hierarchy within each group, are weighted equally, as in Altin (2020). The overall inconsistency ratio was 0.02, which is an acceptable value.

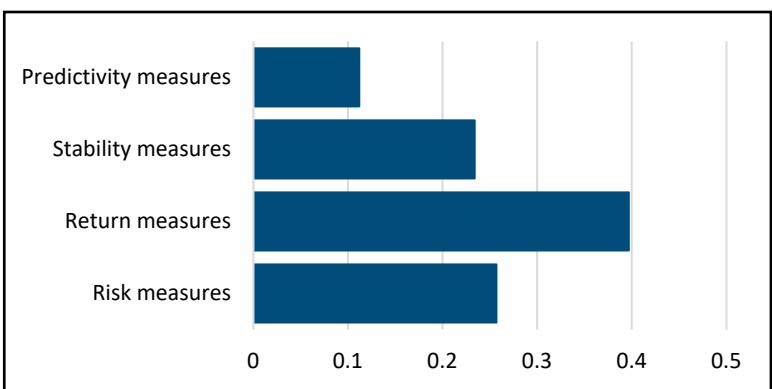

**Figure 2.** Estimation of criteria weights using Expert Choice.

*4.3. PROMETHEE*

In the next phase, six different portfolios were evaluated using the PROMETHEE method. For the implementation of the PROMETHEE method, the mathematical software Visual PROMETHEE was used, which has been used in more than 2400 papers, of which more than 120 were in the field of finance (Mareschal 2020).

First, the preference function for each criterion is determined. This should be determined primarily by the nature of the criterion and the decision-maker's viewpoint. Although some authors, such as Pendaraki and Zopounidis (2003) and Marasović and Babić (2011), use more than one generalized criterion, Turcksin et al. (2011) point out that the PROMETHEE guidelines recommend using a linear preference function for quantitative assessments. Bouri et al. (2002) follow the same practice when financial parameters are involved. After all alternatives have been evaluated by each criterion, an evaluation matrix is constructed (Table 2). Detailed descriptive statistics for all criteria are presented in Appendix B.

**Table 2.** Evaluation matrix.

| Criteria | SD | MAD | CVaR | ML | AR | CMR | SR | CR | OR | CVR | PRED |
|---|---|---|---|---|---|---|---|---|---|---|---|
| Max/Min | min | min | min | max | max | max | max | max | max | max | max |
| Weights | 0.0642 | 0.0642 | 0.0642 | 0.0642 | 0.1985 | 0.1985 | 0.0585 | 0.0585 | 0.0585 | 0.0585 | 0.1120 |
| Naïve | 0.0245 | 0.0316 | 0.0469 | 0.9431 | 0.3433 | 1.0269 | 1.4102 | 4.7509 | 1.0604 | 6.8771 | −0.0016 |
| Mean-Var | 0.0202 | 0.0268 | 0.0418 | 0.9497 | 0.1122 | 0.9994 | 1.2023 | 5.6683 | 1.0739 | 5.9092 | −0.0023 |
| Mean-SR | 0.0253 | 0.0301 | 0.0524 | 0.9370 | 0.2878 | 0.9989 | 1.2507 | 5.7628 | 0.9791 | 6.2522 | −0.0101 |
| Minimax | 0.0224 | 0.0262 | 0.0394 | 0.9488 | 0.3173 | 1.0364 | 1.5004 | 8.4781 | 1.0148 | 9.6449 | 0.0124 |
| Mean-CVaR | 0.0211 | 0.0276 | 0.0426 | 0.9482 | 0.2373 | 1.0485 | 1.1340 | 3.9506 | 1.0752 | 5.5934 | 0.0106 |
| Mean-MAD | 0.0217 | 0.0273 | 0.0433 | 0.9451 | 0.1105 | 1.0022 | 1.1676 | 2.6284 | 1.0429 | 5.1508 | −0.0051 |

Source: The authors.

After determining the evaluation matrix and preference functions, the scenarios are evaluated and ranked using PROMETHEE decision-making software. Thus, minimax is the leader of the portfolio's performance in terms of positive ($\phi^+$) flow, followed by mean-CVaR, Naïve diversification, mean-variance, mean-SR, and mean-MAD. In terms of negative ($\phi^-$) flow, the ranking of alternatives from best to worst performance is: minimax, mean-CVaR, Naïve diversification, mean-variance, mean-MAD, and mean-SR. PROMETHEE II is based on net flows and results in a complete ranking of alternatives. The incomparable status does not exist; thus, these alternatives can be ordered from best to worst. The positive and negative flows (PROMETHEE I) and net flow ($\phi$) values (PROMETHEE II) obtained from this evaluation are displayed in Figures 3 and 4 and Table 3. The results show that the portfolio with the highest average ranking is the best one, while the one with the lowest average ranking is the worst one.

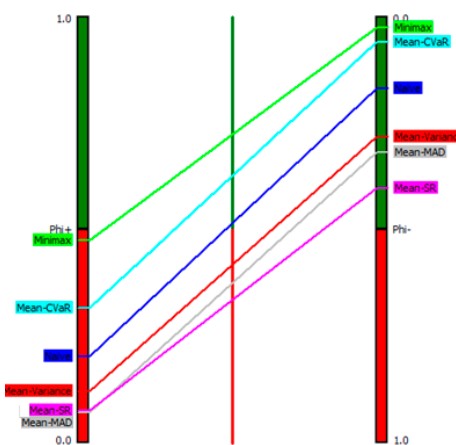

**Figure 3.** Diagram for partial ranking.

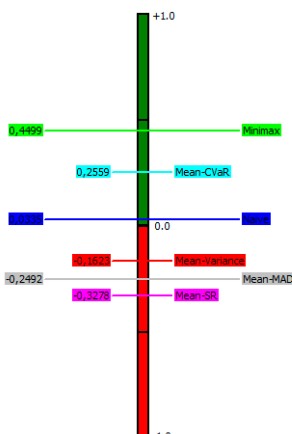

**Figure 4.** Diagram for Complete Ranking.

**Table 3.** PROMETHEE II Complete Ranking.

| Rank | Alternative | Phi | Phi+ | Phi− |
|------|-------------|--------|--------|--------|
| 1 | Minimax | 0.4499 | 0.4745 | 0.0247 |
| 2 | Mean-CVaR | 0.2559 | 0.3159 | 0.0600 |
| 3 | Naïve | 0.0335 | 0.1677 | 0.1677 |
| 4 | Mean-Variance | −0.1623 | 0.1202 | 0.2825 |
| 5 | Mean-MAD | −0.2492 | 0.0698 | 0.3190 |
| 6 | Mean-SR | −0.3278 | 0.0750 | 0.4028 |

Source: The authors.

Furthermore, we disaggregated the computation of net flows for each alternative in detail by PROMETHEE Rainbow diagram, emphasizing the good and weak features of each model (Figure 5). A bar is drawn for each model. The different slices of each bar are colored according to the criteria. Each slice is proportional to the contribution of one criterion (flow value times the weight of the criterion) to the Phi net flow score of the action. Positive (upward) bars correspond to good features, while negative (downward) bars correspond to weaknesses. Models are ranked from left to right according to the PROMETHEE II Complete ranking. The only weakness of the minimax portfolio is the omega ratio, but all other measures have a positive impact on the phi net flow. On the other hand, the positive contribution of the risk measures is the smallest compared with the others.

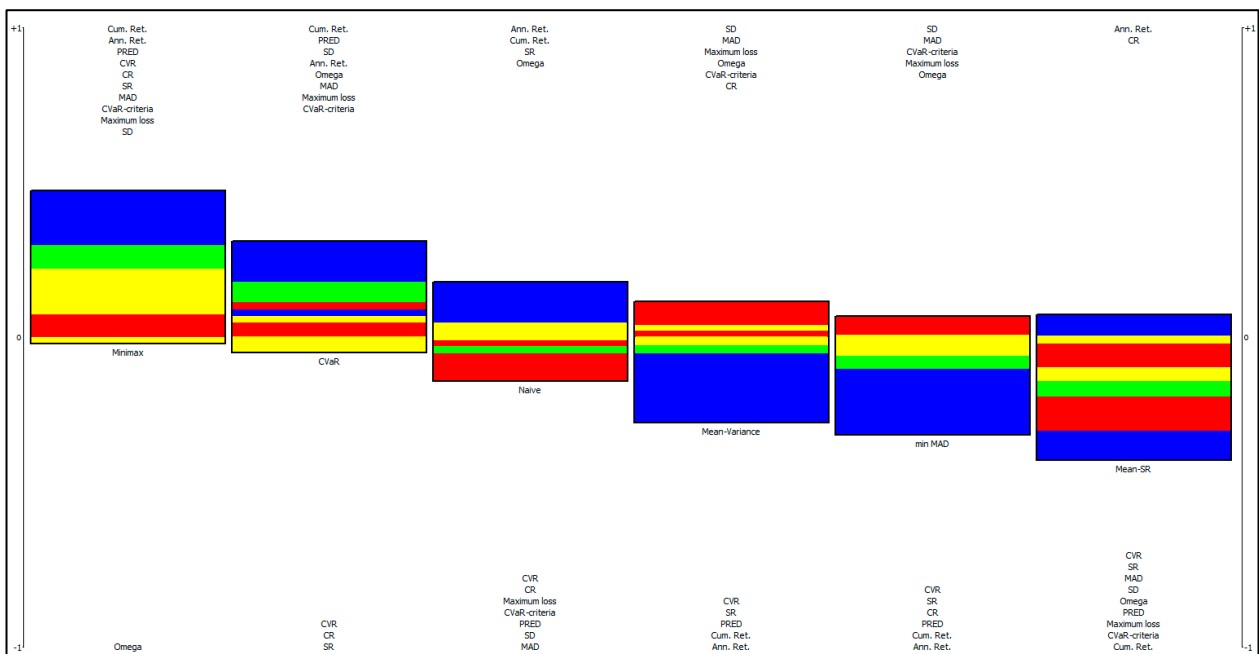

**Figure 5.** PROMETHEE Rainbow Diagram.

Additionally, the decision problem is visualized in the GAIA plane, where the 11-dimensional space of criteria is projected onto a 2-dimensional plane. The plane axes, U and V, represent the latent dimensions of the included criteria obtained by principal component analysis (PCA). In this plane, the alternatives are represented by points and the criteria by vectors. The delta-parameter is 81.20%, which represents the share of the initial total information explained by the dimensions U and V.

Clusters of similar alternatives can easily be detected due to the analysis of the GAIA plane. According to Figure 6, the return, predictability, and stability ratios, except the omega ratio, are close to each other. Similar conclusions are obtained for the risk ratios and the omega ratios that are in the second cluster. In addition to the above analysis, the existence of relationships between criteria within clusters is noticeable in the correlation matrix (Appendix A). A portfolio of minimax is the only alternative that has a positive score on both dimensions U and V, which is expected considering that minimax has excellent performances in most criteria from both clusters.

In the last phase, a sensitivity analysis was carried out, and stability intervals were determined with the aim of validating the proposed strategies and confirming the robustness of the results. Mitkova and Mlynarovič (2019) emphasize that the specific values for the weights of the criteria are still questionable, even in cases where they were assessed by experts, as they depend on the preferences of individual investors. Ikwan et al. (2020) point out that the weights suggested by experts are a good starting point for stability analysis because subsequent changes can be interpreted as reflecting a stronger (or weaker) preference for a particular criterion compared with the value used for the initial ranking. The stability intervals provide an indication of the range over which the particular criterion weights can be varied ceteris paribus without affecting the PROMETHEE II complete ranking or the first position of the minimax portfolio (Table 4).

The minimax portfolio remains first-ranked in the case that even six of the eleven criterion weights can be taken at any value. For all other criteria, the weights can be three to fifteen times higher, and minimax remains first ranked. The easiest way to make a change at the top of the ranking is to increase the weight for the CR criterion by three times. It is clear that this is also unrealistic, since a criterion has a weight of more than 60%. In other words, if the investor's attitude attributes higher importance to any criterion, minimax will

remain in the first position as long as the importance of the criteria to an investor does not exceed 0.27 for OR, 0.48 for SD, 0.61 for CR, etc.

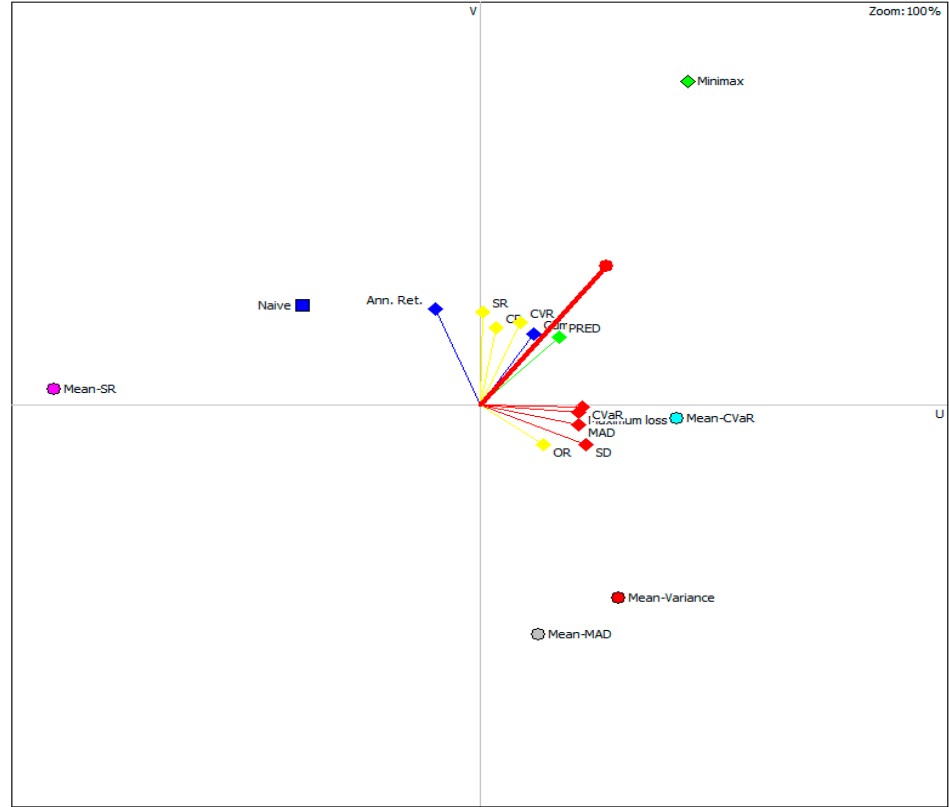

**Figure 6.** GAIA Plane.

**Table 4.** Stability intervals.

| Criteria | Weight | Minimax Remain 1st | Overall Ranking Remain Same |
|---|---|---|---|
| SD | 0.0642 | 0.00–0.48 | 0.00–0.22 |
| ML | 0.0642 | 0.00–0.92 | 0.00–0.36 |
| MAD | 0.0642 | 0.00–1.00 | 0.00–0.21 |
| CVaR | 0.0642 | 0.00–1.00 | 0.00–0.32 |
| AR | 0.1985 | 0.00–0.94 | 0.00–0.26 |
| CMLR | 0.1985 | 0.00–0.61 | 0.00–0.61 |
| SR | 0.0585 | 0.00–1.00 | 0.00–0.25 |
| CR | 0.0585 | 0.00–1.00 | 0.00–0.15 |
| OR | 0.0585 | 0.00–0.27 | 0.00–0.27 |
| CVR | 0.0585 | 0.00–1.00 | 0.00–0.73 |
| PRED | 0.1120 | 0.00–1.00 | 0.00–1.00 |

Source: The authors.

The stability intervals for the change in the overall ranking are somewhat narrower, which is to be expected considering any change between any two positions leads to a change in the overall ranking. The lowest absolute value of a weight that would change the overall ranking is 0.15 for the CR criterion instead of the initial value of 0.0585.

Therefore, the results confirm the findings of Bouri et al. (2002) and Vetschera and de Almeida (2012), who found that the optimal solution does not depend on the weights given to each criterion. On this basis, we can consider our results to be quite robust with respect to the structure of the weights involved.

## 5. Conclusions

This study contributes to the spectrum of MCDM approaches that allow the integration of conventional criteria with other relevant criteria within the portfolio performance evaluation process. The methodology proposed in this paper is based on a combination of AHP and PROMETHEE methods, supported by GAIA and sensitivity analyses. Due to the advantages and disadvantages of both methods, this study applied AHP to structure the decision-making problem and determine the weights of the criteria. PROMETHEE was used for aggregation of criteria, ranking of alternatives, and sensitivity analyses. The presented methodology provides comprehensive support for portfolio performance evaluation based on different risk measures. The empirical results underline the importance and efficiency of successfully engaging multi-criteria methods to find an appropriate balance between different performance measures in the investment decision-making process. Based on the information from PROMETHEE II, GAIA, and the sensitivity analysis, recommendations for the best trade-off can be formulated. Based on the empirical results of the 57-stock sample, the minimax model outperformed the other models in terms of stability, predictability, and return criteria, while mean-CVaR manifested the best performance according to the risk criteria. The proposed methodology could be applied to any portfolio performance comparison. Additionally, the developed method enables investors to choose the best portfolio based on their own preferences and specific market data.

Although this study uses a combined AHP-PROMETHEE approach to rank the portfolio selection models, some limitations have been identified. The main limitation is the subjective choice of criteria. Therefore, special attention should be paid to this issue in future research. In addition, further research should focus on developing a dynamic model for comparing scenarios by year and comparing the results with other MCDM methods.

**Author Contributions:** Conceptualization, M.S., A.A.-B. and A.D.; Methodology, M.S. and A.A.-B.; Software, M.S.; Validation, M.S., A.A.-B. and A.D.; Formal Analysis, M.S.; Investigation, M.S.; Resources, M.S.; Data Curation, M.S.; Writing—Original Draft Preparation, M.S.; Writing—Review & Editing, M.S., A.A.-B. and A.D.; Visualization, M.S. and A.A.-B.; Supervision, A.A.-B. and A.D. All authors have read and agreed to the published version of the manuscript.

**Funding:** This research received no external funding.

**Informed Consent Statement:** Not applicable.

**Data Availability Statement:** The dataset used in this study can be made available upon request.

**Conflicts of Interest:** The authors declare no conflict of interest.

## Appendix A

**Table A1.** Correlation matrix of criteria.

|  | SD | ML | MAD | CVaR | AR | CMLR | SR | CR | OR | CVR | PRED |
|---|---|---|---|---|---|---|---|---|---|---|---|
| SD | 1.00 | | | | | | | | | | |
| ML | −0.81 | 1.00 | | | | | | | | | |
| MAD | 0.86 | −0.67 | 1.00 | | | | | | | | |
| CVaR | 0.85 | −0.97 | 0.74 | 1.00 | | | | | | | |
| AR | 0.29 | −0.17 | 0.24 | 0.17 | 1.00 | | | | | | |
| CMLR | 0.12 | −0.09 | 0.02 | 0.08 | 0.76 | 1.00 | | | | | |
| SR | −0.15 | 0.14 | −0.17 | −0.17 | 0.81 | 0.68 | 1.00 | | | | |
| CR | 0.00 | 0.08 | −0.04 | −0.09 | 0.88 | 0.68 | 0.92 | 1.00 | | | |
| OR | 0.11 | 0.08 | 0.15 | −0.09 | 0.26 | 0.31 | 0.21 | 0.19 | 1.00 | | |
| CVR | −0.04 | 0.08 | −0.08 | −0.11 | 0.88 | 0.72 | 0.94 | 0.98 | 0.20 | 1.00 | |
| PRED | −0.08 | 0.12 | −0.10 | −0.14 | 0.27 | 0.25 | 0.24 | 0.25 | 0.09 | 0.25 | 1.00 |

## Appendix B

**Table A2.** Descriptive statistics for criteria.

|  | SD | MAD | CVaR | ML | AR | CMR | SR | CR | OR | CVR | PRED |
|---|---|---|---|---|---|---|---|---|---|---|---|
| Mean | 0.028486 | 0.033400 | 0.047229 | 0.819443 | 0.229557 | 0.901543 | 1.10339 | 4.47109 | 0.900686 | 5.64087 | 0.016557 |
| Median | 0.022400 | 0.027600 | 0.043300 | 0.945100 | 0.237300 | 1.00220 | 1.20230 | 4.75090 | 1.04290 | 5.90920 | −0.001600 |
| Std. Deviation | 0.015853 | 0.013718 | 0.008588 | 0.333059 | 0.093986 | 0.310632 | 0.479598 | 2.65749 | 0.372970 | 2.86971 | 0.042870 |
| Skewness | 2.574 | 2.540 | 1.557 | −2.645 | −0.278 | −2.624 | −2.210 | −0.299 | −2.599 | −1.064 | 2.450 |
| Kurtosis | 6.707 | 6.556 | 2.236 | 6.996 | −1.634 | 6.911 | 5.417 | 0.713 | 6.807 | 3.068 | 6.198 |
| Minimum | 0.0202 | 0.0262 | 0.0394 | 0.0642 | 0.1105 | 0.1985 | 0.0585 | 0.0585 | 0.0585 | 0.0585 | −0.0101 |
| Maximum | 0.0642 | 0.0642 | 0.0642 | 0.9497 | 0.3433 | 1.0485 | 1.5004 | 8.4781 | 1.0752 | 9.6449 | 0.1120 |

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
