# Peer review of "A Combined AHP-PROMETHEE Approach for Portfolio Performance Comparison"

_ijfs, doi:10.3390/ijfs11010046_

Round 1
Reviewer 1 Report
Report on the paper IJFS " A Combined AHP-PROMETHEE Approach for Portfolio Per-2 formance Comparison"
This paper presents a new method for comparing portfolio performance based on multiple-criteria decision-making (MCDM). The proposed approach combines the Analytical Hierarchy Process (AHP) and the Preference Ranking Organization Method for Enrichment Evaluations (PROMETHEE) to hierarchically evaluate a finite number of alternatives under different criteria. The methodology is applied to compare real portfolios selected on the basis of different risk measures using weekly return data for a sample of stocks that are components of the STOXX Europe 600 index for the period 2000-2020. The results suggest that the simultaneous engagement of different performance measures and the investor’s attitude towards the importance of these measures are notably important in the portfolio efficiency estimation process.
Comments:
1- The approach used in this paper is interesting, the authors show the usefulness of this approach but it would be interesting to orient this motivation towards methods well known in the literature and to show the advantage of this approach compared to other methods.
2- Better define « ELECTRE»
3- The paper considers broadly defined traditional asset classes. I was surprised not to see bond indices in the traditional set.
4- there is no descriptive statistics table to properly analyze the distribution of variables considered in this study
5- Figure 6 is not clear; the axes need to be better defined. it would be better to the characters to be better readable
6- The probability levels considered for the different methods of optimization and risk measurement are not indicated, a sensitivity study with respect to the probability levels may be of interest.
7- « the investor’s attitude » is mentioned in the abstract but in the paper the authors do not address this important element. how can the proposed approach take into account the risk aversion of investors? and is it possible to conduct a sensitivity study of the results with respect to this factor?
8- While the methodology is well crafted to respond to the data type, it is essential to highlight how the research methods incorporate structural differences. Once the authors have addressed the first point, they can indicate how their methodology can be used across the sample.
9- Authors should add Managerial Implications sections and clearly address the research contributions in theory and suggest how the findings may be important for policy, practice, theory, and subsequent research.
Author Response
Response to Reviewer 1 Comments
Point 1: The approach used in this paper is interesting, the authors show the usefulness of this approach but it would be interesting to orient this motivation towards methods well known in the literature and to show the advantage of this approach compared to other methods.
Response 1: Adopted, we have added a sentence regarding the advantages of the used method (AHP-PROMETHEE) compared to other commonly used methods, as follows: This hybrid approach is especially advantageous utilizing the strengths of both individual methods. AHP enables the decomposition of a complex problem into its constituent parts and the de-termination of weights for criteria, while the PROMETHEE method allows the investor to deter-mine the preference function, complete ranking and analysis of the robustness of the results.
Also, we changed part of section 1. Introduction to better explain motivation of using proposed method and contribution of study: “It is not easy to compare models based on different risk measures (Hunjra et al., 2020) because each model dominates in its own risk space (Byrne & Lee, 2004). Accordingly, comparing the performance of different models requires the use of several different indicators (Kalayci et al., 2019). The motivation for this study is developing of a comprehensive evaluation approach that reflects the multidimensionality of performance. Although the integration of AHP and PROMETHEE has been applied to other financial decisions (Ishak et al., 2019), the contribution of this study is introducing of new MCDM approach based on AHP and PROMETHEE for portfolio performance comparison problems.
Point 2: Better define «ELECTRE»
Response 2: Adopted, due to an accidental mistake during the final editing of the text, we forgot to introduce acronym before using it in the text. In this sense, the first time we use the term "ELECTRE", we added: “Elimination and Choice Translating Reality method (ELECTRE)”.
Point 3: The paper considers broadly defined traditional asset classes. I was surprised not to see bond indices in the traditional set.
Response 3: We appreciate the reviewer’s insightful suggestion, but the research included a sample of selected stocks from the composition of the Europe STOXX 600 index. The research is primarily focused on the development and application of a specific methodology focused on the specifics of using different risk measures in the portfolio selection process. Papers that try to compare different models of portfolio selection also performed the selection on equity market, such as for example: https://doi.org/10.1108/14635780410569489; https://doi.org/10.1016/j.frl.2017.07.013; doi:10.3934/jimo.2009.5.33; https://doi.org/10.1016/j.ejor.2017.08.001; https://doi.org/10.1108/JM2-02-2017-0021 etc.
Point 4: There is no descriptive statistics table to properly analyze the distribution of variables considered in this study.
Response 4: We added descriptive statistics table based od evaluation matrix for all alternative and criteria in Appendix). We did not want to burden the main text with descriptive statistics, because evaluation matrix is sufficient for the purpose of applying MCDM methods.
Descriptive statistics for all stock returns and hypothesis testing of the normal distribution of returns (57 stocks) were performed and are available upon request. Considering that the focus of the paper is the development of a new methodology, and not the research of market characteristics, we omitted the discussion of this from the main text. If you think it would contribute to the quality of the discussion, we can insert the results into the main text.
Point 5: Figure 6 is not clear; the axes need to be better defined. it would be better to the characters to be better readable.
Response 5: Adopted, due to an accidental mistake during the final editing of the text, we omitted definition and explanation for axes. We changed part related to GAIA analysis in the section 4.3 as follows: “Additionally, the decision problem is visualized in the GAIA plane, where the 11-dimensional space of criteria is projected onto a 2-dimensional plane. The plane axes, U and V, represent the latent dimensions of included criteria obtained by principal component analysis (PCA). In this plane, the alternatives are represented by points and the criteria by vectors. Delta-parameter is 81.20%, which represent the share of initial total information explained by the dimensions U and V.
Clusters of similar alternatives can easily be detected due to analysis of the GAIA plane. According to Figure 6, return, predictability and stability ratios, except omega ratio, are close to each other. Similar conclusions are obtained for the risk ratios and the omega ratio that are in the second cluster. In addition to the above analysis, the existence of relationships between criteria within clusters is noticeable in the correlation matrix (Appendix A). Minimax based portfolio is the only alternative that has a positive score on both dimensions U and V, which is expected considering that minimax has excellent performances in the most criteria from the both clusters.”
As for the character in the figures, it is an output from the software that cannot be edited. We make the figures as readable as possible.
Point 6: The probability levels considered for the different methods of optimization and risk measurement are not indicated, a sensitivity study with respect to the probability levels may be of interest.
Response 6: We appreciate the reviewer’s insightful suggestion. Considering that the mathematical model of multi-criteria decision-making (AHP-PROMETHEE) was proposed in the study, the methods used were not based on the inclusion of probability in the optimization. The subject of sensitivity analysis are weights. Sensitivity analysis with respect to probability levels is not applicable in this case.
When it comes to the selected portfolio optimization models, the only model that requires the inclusion of probability is CVaR and it is stated: "A confidence interval of 95% was used to calculate the CVaR value."
Point 7: «the investor’s attitude» is mentioned in the abstract but in the paper the authors do not address this important element. how can the proposed approach take into account the risk aversion of investors? and is it possible to conduct a sensitivity study of the results with respect to this factor?
Response 7: We appreciate the reviewer’s insightful suggestion. In this study, «the investor's attitude» does not refer to the classic definition of risk aversion related to the utility function, but it is expressed through the weights of the criteria in the multi-criteria decision-making model. One group of criteria are measures of risk and investors express their attitude towards risk by determining the weighting for these criteria.
In the section 4.2. we added sentence: “These weights reflect the investor's attitude towards the importance of different criteria.”
In the section 4.3. sentence is changed to highlight the meaning of the investor's attitude in this context: “In other words, if the investor’s attitude attributes a higher importance to any criteria, minimax will remain in the first position as long as importance of criteria for an in-vestor don’t exceeds 0.27 for OR, 0.48 for SD, 0.61 for CR, etc.”
Point 8: While the methodology is well crafted to respond to the data type, it is essential to highlight how the research methods incorporate structural differences. Once the authors have addressed the first point, they can indicate how their methodology can be used across the sample.
Point 9: Authors should add Managerial Implications sections and clearly address the research contributions in theory and suggest how the findings may be important for policy, practice, theory, and subsequent research.
Response 8 and 9: The aim of this approach is to contribute to theory by developing a new comprehensive method for portfolio performance comparison. In this sense, we changed part of section 1. Introduction as follows: “It is not easy to compare models based on different risk measures (Hunjra et al., 2020) because each model dominates in its own risk space (Byrne & Lee, 2004). Accordingly, comparing the performance of different models requires the use of several different indicators (Kalayci et al., 2019). The motivation for this study is developing of a comprehensive evaluation approach that reflects the multidimensionality of performance. Although the integration of AHP and PROMETHEE has been applied to other financial decisions (Ishak et al., 2019), the contribution of this study is introducing of new MCDM approach based on AHP and PROMETHEE for portfolio performance comparison problems.”
In Conclusion section, we added: “Additionally, the developed method enables investors to choose the best portfolio based on their own preferences and specific market data.”
Reviewer 2 Report
Dear Authors
I want to report that I have performed the evaluation on this paper, which integrates the AHP method with PROMETHEE for investment portfolio evaluation. After the initial analysis, I verified some points that can be improved:
1) Concerning the citation of the references not according to the MDPI standard, I suggest the authors observe the authors' manual and adjust them;
2) In section 2, the author's report works where multicriteria methods were used to treat problems related to investment portfolios. I suggest the authors reinforce these arguments by including the following articles: https://doi.org/10.36792/rvu.v93i93.43; https://doi.org/10.1108/JM2-02-2017-0021; https://doi.org/10.1504/IJPMB.2022.121599; https://doi.org/10.1142/S021848852250009X.
3) I suggest the authors justify the choice of the AHP method to elicit the weights of the criteria used in the PROMETHEE method. Why didn't the authors choose the ENTROPY or CRITIC methods to obtain the weights of the criteria? I suggest they use https://doi.org/10.3390/electronics11111720 to justify the choice of AHP; it says that AHP is the most used method in problem-solving and integration with other methods.
4) I suggest inserting in the abstract the advantages and disadvantages of this method in relation to the others reported in section 2.
5) Indications of future research should be informed.
Good review.
Best Regards
Reviewer
Author Response
Response to Reviewer 2 Comments
Point 1: Concerning the citation of the references not according to the MDPI standard, I suggest the authors observe the authors' manual and adjust them.
Response 1: Adopted. Once again, we carefully went through all the references and adjusted them with the MDPI standards.
Point 2: In section 2, the author's report works where multicriteria methods were used to treat problems related to investment portfolios. I suggest the authors reinforce these arguments by including the following articles: https://doi.org/10.36792/rvu.v93i93.43; https://doi.org/10.1108/JM2-02-2017-0021; https://doi.org/10.1504/IJPMB.2022.121599; https://doi.org/10.1142/S021848852250009X.
Response 2: Adopted, additional relevant papers are included in the literature review section taking into account the very useful suggestions of all reviewers.
Point 3: I suggest the authors justify the choice of the AHP method to elicit the weights of the criteria used in the PROMETHEE method. Why didn't the authors choose the ENTROPY or CRITIC methods to obtain the weights of the criteria? I suggest they use https://doi.org/10.3390/electronics11111720 to justify the choice of AHP; it says that AHP is the most used method in problem-solving and integration with other methods..
Response 3: Adopted. We appreciate the reviewer's insightful suggestion and recommendation to use this very useful work in argumentation. Certainly, the ENTROPY and CRITIC methods were considered when designing the research methodology. The AHP method was chosen because it is the most commonly used method and because there is well-known software support for its use. Also, the sensitivity analysis drew attention to the importance of the choice of method in a certain way. We have expanded and emphasized the argumentation for using this method by changing this sentence in section 3. Theoretical framework: “Since PROMETHEE does not provide any formal guidelines on how weights can be determined yet, the AHP addresses how to assign the weight of each criterion (), as the most commonly used method for this purpose in hybrid MCDM models (Basílio et al., 2022).”
Point 4: I suggest inserting in the abstract the advantages and disadvantages of this method in relation to the others reported in section 2.
Response 4: Adopted, we added in the abstract the sentence: “This hybrid approach is especially advantageous utilizing the strengths of both individual methods. AHP enables the decomposition of a complex problem into its constituent parts and the determination of weights for criteria, while the PROMETHEE method allows the investor to determine the preference function, complete ranking and analysis of the robustness of the results.”
Point 5: Indications of future research should be informed.
Response 5: We appreciate the reviewer’s insightful suggestion, but in the conclusion indications for future research are given: “Therefore, in future research, special attention should be paid to this issue. Also, further research should be focused on developing a dynamic model for comparing sce-narios by year and comparing results with other MCDM methods.”
Reviewer 3 Report
This paper focuses on portfolio performance evaluation by using a combined AHP-PROMETHEE approach. Methodology has been applied in comparison of real portfolios, selected on the basis of different risk measures by using weekly return data for a sample of stocks that are components of the STOXX Europe 600 index for the period 2000-2020. Experimental results show the performance of the proposed approach. Followings are my concerns:
1. The motivation is not clear. Please explain the importance of the proposed solution.
2. Please highlight your contributions in introduction.
3. To show the importance of the research topic, some related works can be reviewed and added as literature review. For example, Cardinality-constrained portfolio selection via two-timescale duplex neurodynamic optimization.
4. Why 57 randomly selected stocks from STOXX 238 Europe were used in experiment? Please justify.
5. The list of references should be carefully checked to ensure consistency with between all references and their compliances with the journal policy on referencing.
Author Response
Response to Reviewer 3 Comments
Point 1: The motivation is not clear. Please explain the importance of the proposed solution.
Response 1: We appreciate the reviewer's insightful suggestion. The motivation for this study is developing of a comprehensive evaluation approach that reflects the multidimensionality of performance. In that sense, we change part of the section 1. As follows: “It is not easy to compare models based on different risk measures (Hunjra et al., 2020) because each model dominates in its own risk space (Byrne & Lee, 2004). Accordingly, comparing the performance of different models requires the use of several different indicators (Kalayci et al., 2019). The motivation for this study is developing of a comprehensive evaluation approach that reflects the multidimensionality of performance. Although the integration of AHP and PROMETHEE has been applied to other financial decisions (Ishak et al., 2019), the contribution of this study is introducing of new MCDM approach based on AHP and PROMETHEE for portfolio performance comparison problems.”
Also, we added in the abstract the sentence “This hybrid approach is especially advantageous utilizing the strengths of both individual methods. AHP enables the decomposition of a complex problem into its constituent parts and the determination of weights for criteria, while the PROMETHEE method allows the investor to determine the preference function, complete ranking and analysis of the robustness of the results.”
Point 2: Please highlight your contributions in introduction.
Response 2: Adopted, contribution is further highlighted in introduction by adding the following part: Although the integration of AHP and PROMETHEE has been applied to other financial decisions (Ishak et al., 2019), the contribution of this study is introducing of new MCDM approach based on AHP and PROMETHEE for portfolio performance comparison problems.
The contribution is also highlighted in the section 3. Theoretical framework: “This section provides a brief description of the multi-criteria optimization methods: AHP and PROMETHE II, which are analyzed in this paper. An approach combining AHP and PROMETHEE methods for ranking alternatives has already been used in various fields (Turcksin et al. 2011 and Komchornrit 2021), but this is the first time that AHP and PROMETHEE have been used as a hybrid model comparing different models portfolio performance.”
Point 3: To show the importance of the research topic, some related works can be reviewed and added as literature review. For example, Cardinality-constrained portfolio selection via two-timescale duplex neurodynamic optimization.
Response 3: Adopted, additional relevant papers are included in the literature review section taking into account the very useful suggestions of all reviewers.
Point 4: Why 57 randomly selected stocks from STOXX 238 Europe were used in experiment? Please justify.
Response 4: We appreciate the reviewer's insightful suggestion. We are aware that the issue of the number of stocks in the sample is a very sensitive issue. In a very large number of papers that we have analyze and which are focused on the development of methodology for performance evaluation, the discussion on this issue is neglected and the number of shares is taken randomly, without any explanation. The logic we followed in this paper is explained in the section: “The number of stocks in the sample follows the practice of Lee & Gankhuyag (2020), and maintains the fact that less than 60 stocks are needed to realize the full benefits of diversification and consequently the risk is reduced to the level of market risk (Raju & Agarwalla, 2021).” In this way, the number of selected stocks in the initial sample is higher than in most similar works, which we considered acceptable. After we selected 60 stocks, 3 stocks were omitted due to missing data (asynchronous operations). Considering that the focus of the paper is the development of a new methodology, and not the research of market characteristics, we did not increase the sample or burden the text with detailed explanations.
Point 5: The list of references should be carefully checked to ensure consistency with between all references and their compliances with the journal policy on referencing.
Response 5: Adopted. We carefully went through all the references once again and adjusted them with the MDPI standards.
Round 2
Reviewer 2 Report
Dear Authors
With the usual compliments, I congratulate you for the current text and the implementations made in accordance with the reviewers' suggestions, thus improving the gaps observed in the first revision round. I consider my suggestions taken into account, with no further comments.
Best regards
Reviewer